# Hypersensitivity Reactions to Iodinated Contrast Media

**DOI:** 10.3390/biomedicines10051036

**Published:** 2022-04-30

**Authors:** Tsu-Man Chiu, Sung-Yu Chu

**Affiliations:** 1Department of Dermatology, Changhua Christian Hospital, Changhua City 50073, Taiwan; 68003@cch.org.tw; 2Institute of Medicine, Chung Shan Medical University, Taichung 40201, Taiwan; 3Department of Medical Imaging and Intervention, Chang Gung Memorial Hospital, Linkou Branch, Taoyuan 33305, Taiwan

**Keywords:** hypersensitivity, allergy, iodinated contrast media, skin test, basophil activation test, lymphocyte transformation test, drug provocation test, premeditation

## Abstract

At present, iodinated contrast media (ICM) are mostly non-ionic, have low osmolality, and are safe. Even if adverse drug reactions (ADRs) occur, most are chemo-toxic symptoms and require only observation or H1 antihistamines. However, rare, unpredictable, and even life-threatening hypersensitivity can still occur. The aim of this review is to summarize the issues that all relevant staff need to know about and be able to respond to. The most significant risk factor for ICM hypersensitivity is a history of ICM hypersensitivity. For high-risk populations, we must cautiously weigh the advantages and disadvantages of premedication and be aware that breakthrough reactions may still occur. The best policy for patients with a history of severe ICM hypersensitivity is to avoid the same ICM. If ICM are inevitable, skin tests, in vitro tests, and drug provocation tests may help to find a feasible alternative that is safer. The severity of the hypersensitivity is correlated with the positivity rate of these tests, so there is no need for further investigations for patients with only mild reactions. We should also keep in mind that even excipients in ICM may induce hypersensitivity. Detailed, standardized documentation is essential for correct diagnosis and the prevention of future occurrence.

## 1. Introduction

In medical imaging, contrast media can help to distinguish between different tissues in the body and improve the accuracy of disease diagnosis [1]. Different contrast media are available for various medical imaging technologies, such as sonography, magnetic resonance imaging, and X-ray-based imaging (e.g., angiography, computed tomography (CT), and upper-gastrointestinal series studies) [2]. However, the administration of contrast media is associated with adverse drug reactions (ADRs). The incidence of ADRs is higher for contrast media used in X-ray-based imaging than that for sonography and magnetic resonance imaging [2].

X-ray-based contrast media include barium sulfate, iodinated contrast media (ICM), and carbon dioxide (CO_2_). Barium sulfate is only used in fluoroscopic examinations for the gastrointestinal tract, with seldom ADRs [3,4]. For patients with impaired renal function or a history of hypersensitivity to ICM, CO_2_ can be used as a contrast medium in angiography or venography [5].

ICM can be divided into oil-soluble and water-soluble ICM. Oil-soluble ethiodized oil (lipiodol) is limited to use in hysterosalpingography, lymphangiography, and transarterial chemoembolization [6], and has an extremely low incidence of hypersensitivity. However, water-soluble iodinated contrast media also play an important role in the diagnosis of disease and evaluation of the therapeutic response, especially in CT imaging of vessels and tumors. Thus, this review focuses on the hypersensitivity of this type of ICM.

## 2. Classification of ICM

The first generation of ICM was ionic ICM, which has one benzene ring monomer containing three iodine atoms and a side chain with a carboxylic acid (–COOH) group [7]. The osmolality of ionic ICM is 5–7 times that of normal serum. Ionic ICM has been classified as a hypertonic and high-osmolar ICM. The second generation of ICM was non-ionic ICM, which also has one benzene ring monomer with various side chains containing polar alcohol (–OH) groups, but no –COOH groups. Due to its non-ionic characteristics, the osmolality is decreased to 2–3 times that of normal serum, but its radiopacity remains similar. Compared to the osmolality of ionic ICM, monomeric non-ionic ICM are classified as hypotonic or low-osmolar ICM.

Another new type of non-ionic ICM is dimeric ICM, which contains two covalently bound tri-iodinated benzene rings. The osmolality of this non-ionic dimeric ICM is similar to that of normal serum, and non-ionic dimeric ICM are classified as isotonic or iso-osmolar ICM [8].

The radiopacity of ICM is dependent on the iodine concentration (mg/mL), with higher iodine concentration corresponding with higher osmolality. In general, ionic monomeric ICM has higher osmolality than non-ionic monomeric ICM. The different types of ICM and their characteristics are summarized in Table 1.

The ICM solution contains not only contrast media, but also excipients. Sodium calcium edetate is an excipient of both ionic and non-ionic ICM. Tromethamine (trometamol, tris) can be found in the different monomeric and dimeric non-ionic ICM. These excipients may play a role in ICM hypersensitivity [9,10,11]. The most common method for administration of ICM into the body is intravascular injection. The injection speed varies depending on the purpose of imaging and ranges from slow manual injection to injection at 20 mL/s using a power injector.

## 3. Classification of ADRs of ICM

The ADRs related to the intravascular administration of ICM can be divided into two types: hypersensitivity (allergic-like) events and chemo-toxic events [2,4,12]. According to the ACR Manual on Contrast Media [2], ADRs can also be divided into three grades: “mild” (signs and symptoms are self-limited without evidence of progression), “moderate” (signs and symptoms are more pronounced and commonly require medical management), and “severe” (signs and symptoms are often life-threatening and can result in permanent morbidity or death if not managed appropriately). Some mild and moderate reactions have the potential to become severe if not treated.

The detailed symptoms and signs of different grades are summarized in Table 2. Most ADRs of ICM are immediate (acute) with onset within 1 h and commonly within 30 min after intravenous administration of ICM [2,13]. About 70% of them occur within 5 min [14]. Non-immediate (delayed) cutaneous reaction usually has a latency of hours to one week after receiving the ICM [2]. Nearly all life-threatening ADRs occur within 20 min after intravascular administration of ICM. Chemo-toxic reactions to ICM are frequently dependent on the dose and concentration, while hypersensitivity is likely independent of them [2]. Nausea, vomiting, or flushing are occasionally difficult to classify as hypersensitivity or chemo-toxic events.

## 4. Epidemiology

Several studies have revealed a higher incidence of ADRs with ionic ICM (4.17–12.66%) than non-ionic ICM (0.69–3.13%) [14,15,16]. Therefore, non-ionic ICM have mostly been used in X-ray-related studies in recent years, especially in CT and angiography. Standardized classification systems for ADRs of ICM have been established to differentiate the hypersensitivity and chemo-toxic events [2,4], and the incidence of hypersensitivity to different non-ionic ICM has been reported as being 0.37–0.99% [17]. The potential risks of ADRs are allergy, asthma, cardiac status, anxiety, age, and gender [2,18,19]. No external warming of ICM [20], injection dose [21,22,23], injection speed [21,22,23], and family history of hypersensitivity to ICM [17] are also risk factors.

## 5. Phenotype of Hypersensitivity

Hypersensitivity to ICM is divided into immediate hypersensitivity reaction (IHR), in which reactions occur within 1 h of ICM administration, and non-immediate hypersensitivity reaction (NIHR), which occurs more than 1 h later [24,25,26,27]. The latency of most NIHRs is 3 h to 2 days, and they usually resolve in one week. In patients receiving non-ionic ICM, the incidence of IHR and NIHR is about 0.5–3% [25,28]. The manifestations of IHR include urticaria, angioedema, vomiting, abdominal pain, diarrhea, dyspnea, bronchospasm, and drop in blood pressure [24,25,26,27]. The most severe form of IHR is life-threatening anaphylactic shock with hypotension and loss of consciousness, which has a prevalence of 0.04–0.28% [29].

The pathogenesis of most ICM-induced hypersensitivity is unclear. Multiple possible pathomechanisms result in the activation of effector cells and the release of mediators. Histamine release occurs when patients develop urticaria. In the case of IHR, the pathophysiological explanations include the activation of mast cells and basophils (Figure 1), in addition to the release of histamine, tryptase, and other mediators [30]. The activation and release of mediators can occur through the IgE-mediated immune pathway and non-specific pathways, such as activation of the complement systems, activation of the XII clotting system (leading to the production of bradykinin and conversion of L-arginine into nitric oxide) [31], and formation of “pseudoantigens” [6]. A possible underlying specific IgE-mediated mechanism may be responsible for some IHRs of ICM because some individuals have positive skin test results [32]. A recent mouse study showed that iodixanol mechanistically increases the degranulation of mast cells, and more inflammatory factors are synthesized through activation of the PLC-γ and PI3K-related pathways [33].

The most common type of NIHR is maculopapular exanthema (MPE), which accounts for more than 50% of affected individuals [34]. Non-ionic dimeric ICM induces more NIHR than non-ionic monomeric ICM [35]. Other types of NIHR are rare but more severe and even potentially fatal, such as Stevens–Johnson syndrome, toxic epidermal necrolysis, drug reaction with eosinophilia and systemic symptoms, acute generalized exanthematous pustulosis, symmetrical drug-related intertriginous and flexural exanthema, and fixed drug eruption [24,25,27,36,37,38,39,40,41]. Delayed urticaria and angioedema may sometimes appear. NIHR is T cell-mediated, and a skin biopsy may reveal CD4+ and CD8+ T cell infiltration of the skin lesions, whereas a delayed-reading skin test (such as an intradermal test (IDT) or patch test) may show a positive reaction.

The latency of T cell-mediated delayed-type hypersensitivity to ICM ranges from hours to days after administration of the ICM. Usually, drug reactions with eosinophilia and systemic symptoms develop after exposure to the culprit drug for 2 to 4 weeks, but if induced by ICM, the latency is shorter (usually within the week following ICM administration) [39]. A positive lymphocyte transformation test (LTT) and ICM-specific T cell clones have been demonstrated in some patients [42]. The classification and recommendations for ADRs of ICM are shown in Figure 2.

## 6. Diagnosis

When dealing with a patient who has suspected hypersensitivity to a drug, food, or ICM, it is most important to obtain a detailed clinical history. To confirm the severe form of IHR, anaphylaxis, it is suggested to check serum levels of the mediators histamine and tryptase, which are released from mast cells and basophils after activation. The checks should be performed immediately after the event (within 4 h) and 24 h later (baseline). The elimination half-lives of histamine and tryptase are 15 to 20 min and 90 min to 2 h, respectively. Checking the serum level of tryptase is more practical because histamine degrades quickly and is more complicated to measure [30]. Serum levels of tryptase are positively correlated with the severity of IHR, and it is a valuable biomarker that supports the diagnosis of anaphylaxis. A clinically significant rise is defined as a serum tryptase level greater than one of the following: ([1.2 × baseline tryptase] + 2) μg/L, ([baseline tryptase] + 3) μg/L, 1.35 × baseline tryptase, or more than 11.4 μg/L. Among these options, the first has the greatest Youden’s index, which means it has the most significant clinical practicability [44].

The most important criterion for testing ICM hypersensitivity is a history of severe hypersensitivity, either immediate or non-immediate. Although the sensitivity of in vivo and in vitro tests of ICM allergy correlates with the severity of the hypersensitivity [45], a skin test may be positive in cases of mild reactions if performed within a maximum of 6 months from the reaction. Hence, a skin test can be performed in all patients with a history of hypersensitivity reactions if feasible. However, the significance of both tests is limited due to unsatisfactory sensitivity and specificity, as shown in Table 3. In vitro tests are preferred for severe and even life-threatening hypersensitivity reactions such as anaphylaxis.

Once the IHR event is resolved, a skin test can be performed to identify whether the IHR is IgE-mediated. The sensitivity range of skin tests in IHR is 4.2 to 73% and correlates with the severity of the phenotype of IHR [46,47,48,49,50,51,52,53]. In patients with IHR to ICM, pooled per-patient positivity rates of skin tests were 17% (95% CI, 10–26%) in a meta-analysis from 2015, but the positivity rates were increased to 52% (95% CI, 31–72%) in patients with severe IHR. The pooled per-patient positivity rates of skin tests in NIHR patients were 26% (95% CI, 15–41%). Among the skin tests, the rate for the skin prick test (SPT) is 7% (95% CI, 1–30%), the rate for IDT is 22% (95% CI, 13–34%), and the rate for the patch test is 16% (95% CI, 15–41%). If we combine IDT with the patch test, the positivity rate is increased [50].

The cross-reactivity of ICM in skin tests in the IHR phenotype population is 68% (95% CI, 48–83%), and that in the NIHR phenotype population is 39% (95% CI, 29–50%). The specificity of SPT for ICM-induced IHR is about 94.6%, and that of IDT is 91.4–96.3% [51,52]. The negative predictive value of skin tests is 93% (95% CI, 86–96%) [50]. The sensitivity and specificity of skin tests in ICM-induced NIHR are 72% and 96%, respectively, but the skin test is usually performed long after the hypersensitivity event, which decreases the positivity rate [12,48,54,55].

According to these findings, even with negative skin-test results for possible alternative ICM, physicians should pay attention to the patient because recurrence of hypersensitivity reaction may occur. In another study, there was no recurrence of severe hypersensitivity reactions in patients receiving alternative ICM with a negative skin test [56]. Hence, skin tests are invaluable for the investigation of possible alternative ICM, with negative predictive values of 94.2% (95% CI, 89.6% to 97.2%) and 86.1% (95% CI, 72.1–94.7%) for IHR and NIHR, respectively [57]. The skin test should be performed using a panel of many kinds of ICM, including the suspected culprit ICM.

SPT is performed with undiluted ICM (300–320 mg/mL), and then IDT is performed with ICM at dilutions of 1:1000 to 1:10 if the SPT result is negative [27,52]. It is notable that lower concentration of the tested ICM corresponds with lower positivity rates of reactions [53]. IDT is read at 20 min for IHR. Delayed reading of IDT after 48 and 72 h is necessary for delayed-type hypersensitivity, and a positive reaction should present wheals and possibly erythema instead of wheals only, which may be due to osmolality [42]. In severe hypersensitivity reactions, the investigations should be performed very cautiously. For NIHR, a safe stepwise approach may be started with a patch test, followed by delayed-reading IDT with a 1:10 dilution of ICM, and finally, delayed-reading IDT with undiluted ICM (1:1) [50,57,58,59].

Patch tests are performed in delayed-type hypersensitivity and read after 48 h and 96–120 h. However, these tests are more difficult and expensive [52,60]. In fixed drug eruption, PD should be tested in situ on previous lesion sites to avoid false-negative results [34]. The reason for this is that drug-specific resident-memory T cells are present in previously fixed drug eruption lesions [61]. The allergological work-up should be performed with a latency of 1 and 6 months from the latest ICM reaction, or else the sensitivity can be low [46]. The reason for the limited duration of skin reactivity is considered to be IgE clearance [62]. The time interval between the skin test and the immediate hypersensitivity event is 3 months and 48 months in groups with positive and negative skin-test results, respectively [46].

Once the hypersensitivity has resolved, in vitro tests for identifying the culprit ICM are indicated for patients with high risk or severe hypersensitivity phenotype, in addition to cases when a skin test is not available [49]. The basophil activation test (BAT) has been considered to be a useful tool in identifying culprit drugs or ICM in cases of IHR, and its correlation with the skin test and drug provocation test is good [48]. BAT is especially useful for severe forms of IHR such as anaphylactic shock, for which skin tests and drug provocation test are contraindicated. One should also be aware that many factors influence the performance of BAT, such as the selection of stimulants, designed protocol, gating strategies, and cutoff value [48,63]. The sensitivity and specificity of BAT are 46–63% and 89–100%, respectively, depending on the threshold chosen [63,64,65].

The lymphocyte transformation test (LTT) detects and measures the proliferation of circulating lymphocytes specific to the antigen of the culprit drug or ICM upon stimulation by the antigen. The sensitivity and specificity of LTT differ for different antigens. LTT is useful for detecting culprit ICM in patients with NIHR, but a negative result may not rule out the cause [27,66,67]. The sensitivity of LTT ranges from 13 to 75% [65]. It is suggested that LTT be performed at 4–6 weeks after resolution of the event and the withdrawal of corticosteroids and other immunosuppressants, but no more than 2 to 3 years after the reaction [68,69]. Corticosteroid (more than 0.2 mg/kg of prednisolone) and equivalent immunosuppressants may interfere with the performance of LTT.

There is controversy regarding the need for a drug provocation test (DPT), which is considered to be the last step of the diagnostic algorithm. Even if there is a negative result in pretesting, recurrence of the hypersensitivity reaction following re-exposure to ICM may occur [70]. Administration of ICM is potentially harmful, so many contraindications should be noted before performing DPT, such as impaired renal function, current medications with renal toxicity, pregnancy, lactation, thyrotoxic crisis, lactic acidosis, and patients with severe IHR or NIHR [46].

Currently, no standard protocol for drug provocation has been established, such as a graded challenge or full-dose challenge test. The dose of ICM used in DPT ranges from 49 to 100 mL. The negative predictive value of DPT at a Spanish center was 97.3 and 80% when using 10 mL and 50 mL of ICM, respectively [71]. Two protocols are currently used: one involving 5, 15, 30, and 35 mL at 45-min intervals; and another involving 0.05, 0.5, 1, 5, 7.5, 10, and 25 mL at 30-min intervals [72].

A consensus on protocols should be established and validated. Drug provocation tests must be performed by well-trained experts in well-equipped centers for possible emergency treatment. DPT should only be performed for patients with negative skin test, basophil activation test, and lymphocyte transformation test results, or when there is no alternative diagnostic tool and the diagnosis is uncertain [73]. The recommended in vivo and in vitro tests for IHR and NIHR are illustrated in Figure 2.

The excipient tromethamine is a buffering agent and is contained in many cosmetics, drugs, and COVID-19 vaccines. It may cause contact dermatitis, allergy, and even anaphylaxis [9,74]. Another excipient in ICM is sodium calcium edetate. There is one report of a patient with allergy to sodium calcium edetate, who was identified through investigation of the reactions to ICM [10]. Investigations to determine the culprit ICM should include the excipients. Hypersensitivity reactions to excipients are immediate; hence, BAT is performed to identify if excipient or ICM per se is the real culprit. If hypersensitivity to tromethamine or sodium calcium edetate is proven, the patient should be informed to avoid drugs or cosmetics containing the ingredient in the future. Furthermore, pharmaceutical companies should put every effort into developing ICM that contain promising stabilizing agents with reduced immunogenicity.

## 7. Treatment

The reactions after injection of ICM range from mild chemo-toxic reactions to severe life-threatening situations. Most of the adverse reactions to low-osmolality ICM are mild and only require observation or oral H1-antihistamine. Although life-threatening anaphylaxis is rare and unpredictable, remaining vigilant is very important. Staff such as radiologists, technologists, and nurses should be familiar with standard operating procedures for all kinds of reactions. When adverse reactions occur, personnel should notify the supervising radiologist, monitor the vital signs of the patient, administer certain medications, and call for assistance (such as from the emergency department). All emergency equipment and medication should be prepared.

H1 antihistamines are sufficient for mild hypersensitivity symptoms such as itching, but for life-threatening anaphylaxis, intramuscular epinephrine is necessary, and many radiologists are unaware of this [75]. The first-line treatment is 0.01 mg per kilogram of body weight to a maximum of 0.5 mg of epinephrine at a concentration of 1:1000, which should be injected intramuscularly in the lateral aspect of the thigh [76]. When patients manifest airway, breathing, and circulation symptoms, staff should assess and manage them immediately and adequate oxygen supply is critical.

The most common type of NIHR is maculopapular exanthema [34]. NIHR usually has mild to moderate severity and is usually self-limiting, with most cases requiring little or no therapy. However, severe NIHR may occur, such as acute generalized exanthematous pustulosis, drug reaction with eosinophilia and systemic symptoms, Stevens–Johnson syndrome, and toxic epidermal necrolysis. Such cases usually need to be referred to a specialist for treatment. In these circumstances, systemic corticosteroids and even admission are usually needed.

## 8. Prescreening, Prevention and Rapid Drug Desensitization

A questionnaire before the procedure may help to determine whether patients have higher risk of a hypersensitivity reaction. Apart from this, no in vitro or in vivo test has shown good evidence for prescreening populations with high risk. A recent prospective study was performed for the validation of prescreening IDT to predict ICM hypersensitivity, but the results revealed extremely low sensitivity and a low positive predictive value [77]. In patients with a history of hypersensitivity reaction, the intradermal skin test has good negative predictive value [78]. In conclusion, prescreening with skin tests is not recommended due to the extremely low sensitivity and low positive predictive value.

In patients with a hypersensitivity history to their first-choice treatment, rapid drug desensitization is a therapeutic approach which temporarily builds tolerance to the drug in a short period of time. The principle of desensitization is to gradually eliminate the IgE activation of mast cells through small doses of drugs. This is done by starting with a small dose of the drug and slowly increasing the dose until achieving the therapeutic dose needed or the maximum tolerated dose. Rapid drug desensitization has been practiced for many years in taxanes for cancer patients and biologics such as rituximab, and there is a great clinical need. With rapid drug desensitization, the chance of breakthrough reaction is usually less than 10%, and this is usually a mild reaction. Premedication is also included in some desensitization protocols. Despite the long-existing therapeutic approach, there is no evidence to support such drug desensitization in radiocontrast agent hypersensitivity [78,79].

## 9. Recommendations

For patients with a history of hypersensitivity to ICM, the recommendations range from strict avoidance of ICM to imaging alternatives that do not use ICM, premedication, alternative ICM, and the drug provocation test [12,25,80]. Avoidance of imaging studies with ICM is recommended if alternative imaging techniques such as magnetic resonance imaging are applicable [25,27]. Although systemic corticosteroids and antihistamines have been used widely for a long time as premedication, there is no gold-standard regimen of premedication for preventing hypersensitivity reactions to ICM. Breakthrough reactions develop despite premedication.

The American College of Radiology recommends changing an alternative ICM in patients with a prior allergic-like reaction [2]. In patients using alternative ICM with a negative skin test for the ICM, the recurrence rate of IHR is 7% (95% CI, 4–14%), and that of NIHR is 35% (95% CI, 19–55%) [50]. In patients with a history of mild hypersensitivity reaction, the recurrence rate of similar mild reactions is 31.1% with the same ICM, 12% with another ICM, and 7.6% with antihistamine premedication [81,82,83]. In patients with mild ICM-related IHR limited to the skin, the rate of severe IHR after re-administration of the ICM, such as anaphylaxis, is below 1% [84].

The American College of Radiology recommends premedication for patients with prior allergy-like reactions to ICM or for populations at risk [2]. The purpose of corticosteroids is to mitigate the likelihood of hypersensitivity in high-risk individuals or in patients with a history of hypersensitivity to ICM. In the era of high-osmolality ICM administration, premedication may be beneficial for reducing the likelihood of IHR [85]; however, now that high-osmolality ICM are no longer used, the efficacy of premedication remains questionable.

Premedication with systemic corticosteroids and antihistamines may prevent the occurrence of mild to moderate IHR to ICM in circumstances where ICM administration is needed, but the breakthrough reaction rate is 3–4 times that of the population without a history of hypersensitivity [27,81,84]. In patients with a history of severe hypersensitivity reactions, such as anaphylaxis, use of the same ICM should not be attempted, even with premedication. Imaging without ICM is suggested. If the patient must undergo imaging with ICM, prescreening to find a safer alternative is suggested.

In general, avoiding the known culprit ICM is more helpful than premedication with corticosteroid and antihistamines in reducing the recurrence of IHR to ICM [81,83,86]. Premedication is not effective in patients with a history of NIHR to ICM [39,87]. In practice, the most common protocol of oral corticosteroids involves several time points (e.g., 50 mg of prednisone orally at 13, 7, and 1 h before administration of ICM) and H1 antihistamine (e.g., 50 mg of diphenhydramine 1 h before administration of ICM) [88]. Another study suggests performing prophylaxis using 5 h of intravenous corticosteroid (200 mg of hydrocortisone administered at 5 and 1 h, with 50 mg of diphenhydramine administered 1 h before the imaging study). The results suggest that this approach is not inferior to an oral regimen [89]. Nevertheless, premedication is not a panacea, and the radiology team must be prepared for treating breakthrough reactions.

There are many drawbacks of premedication with corticosteroids and antihistamines, such as transient leukocytosis, asymptomatic hyperglycemia, increased risk of infection, lengthened hospitalization, and drowsiness. Furthermore, it may cause more indirect suffering than the direct distress that it is meant to prevent [90]. The use of a substitute for the known ICM is more helpful in reducing the possibility of breakthrough reactions. In summary, no known precautional measures can guarantee against repeat reaction after re-administration of the culprit ICM, and avoiding the known ICM is more helpful than premedication in reducing the recurrence of hypersensitivity to ICM. The recommendations for IHR and NIHR of ICM are demonstrated in Figure 2. A history of chemo-toxic reactions (such as nausea, vomiting, or flushing) does not increase the risk of future ICM-related hypersensitivity, and premedication is not recommended in such cases.

## 10. Standardization of the Documentation of ICM-Induced Hypersensitivity

It is critical to standardize documentation with a detailed description of ICM-induced hypersensitivity for correct diagnosis and further investigations. Staff should avoid using inaccurate terms such as “iodine allergy,” which may interfere with judgment. The suggested standardized documentation should include the following [91]:The precise name of the injected culprit ICM and its dose (volume, concentration);The manifestations of the reaction (e.g., itching, generalized urticaria, dizziness, drop of blood pressure, tachycardia, etc.) to establish the severity of the reaction;The chronology of the adverse reaction;Specific treatment for the ICM-induced hypersensitivity.

Correct medical records lead to logical thinking about the adverse event and may help to prevent recurrence of the same reactions.

## 11. Conclusions

All ICM used in practice are relatively safe but have a possible risk of hypersensitivity, even when used correctly as prescribed. In general, however, the benefits of performing the procedure with ICM when needed outweigh the risks. In patients with a history of hypersensitivity, using an alternative ICM is preferred to using the culprit ICM with premedication because breakthrough reactions may occur. Skin tests (SPT, IDT, PT) and in vitro tests (BAT and LTT) may help to confirm the causality of ICM and hypersensitivity reactions, and help to find a safer alternative ICM. Standardized documentation of the hypersensitivity event leads to better clarification and may help with future research on ICM hypersensitivity.

## Figures and Tables

**Figure 1 biomedicines-10-01036-f001:**
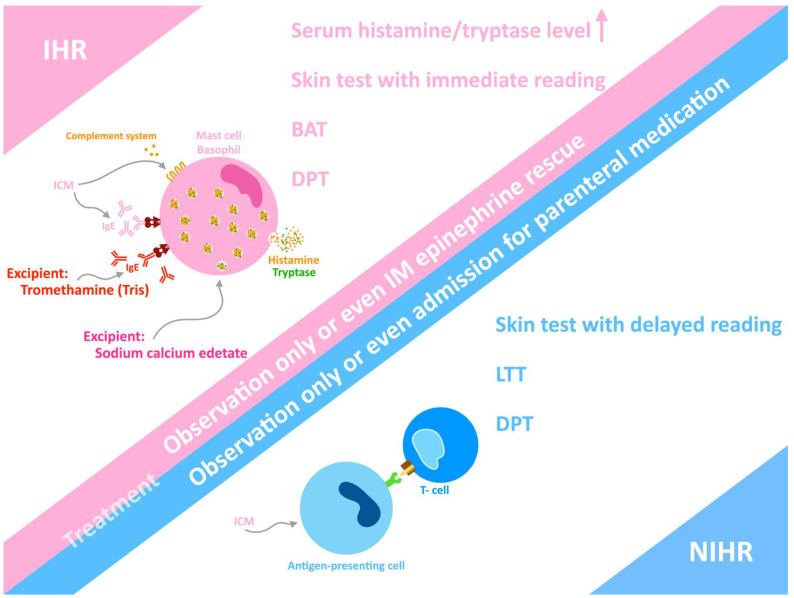
Summary of iodinated contrast media (ICM)-induced hypersensitivity. ICM or excipients may activate the basophil or mast cell to release histamine and other mediators, and then induce immediate hypersensitivity reaction (IHR) via IgE-mediated or non-IgE pathways, such as the complement system pathway. Non-immediate hypersensitivity reaction (NIHR) induced by ICM can be evoked by a T-cell mediated pathway. Skin test, basophil activation test (BAT), drug provocation test (DPT), and lymphocyte transformation test (LTT) can achieve the diagnosis of ICM hypersensitivity. In patients with anaphylaxis or severe, prolonged symptoms of NIHR, prompt and appropriate intervention is needed.

**Figure 2 biomedicines-10-01036-f002:**
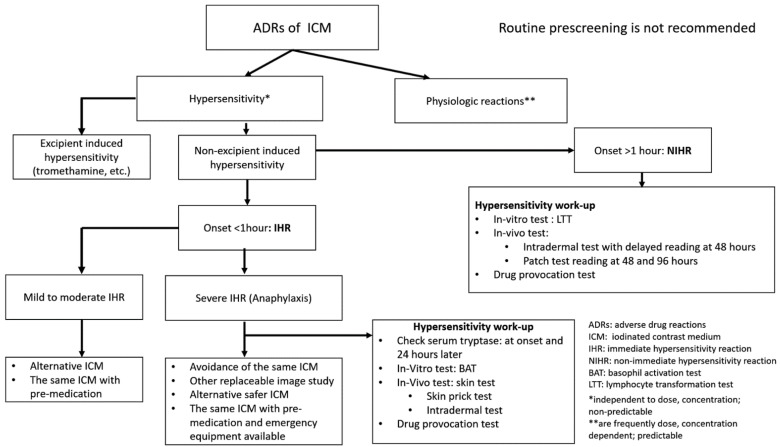
Algorithm for classification and recommendations for ADRs of ICM. For excipient-induced hypersensitivity, investigations can be performed as in IHR work-up. The algorithm is the authors’ proposal and modified with references [25,43].

**Table 1 biomedicines-10-01036-t001:** Types of iodinated contrast media and their characteristics.

Generic Name	Trade Name	Ionic/Non-Ionic	Monomer/Dimer	Iodine Content (mg/mL)	Osmolality (mOsm/kg H_2_O)	Osmolar	Excipients with Reported Hypersensitivity
Iothalamate	Conray	Ionic	Monomer	325	1843	High	Sodium calcium edetate
Amidotrizoate	Urografin 76%	Ionic	Monomer	370	2100	High	Sodium calcium edetate
Ioxithalamate	Telebrix 30	Ionic	Monomer	300	1710	High	Sodium calcium edetate
Diatrizoate	Hypaque 50	Ionic	Monomer	300	1550	High	Sodium calcium edetate
Ioxaglate	Hexabrix	Ionic	Dimer	320	580	Low	Sodium calcium edetate
Iopromide	Ultravist 370	Non-ionic	Monomer	370	774	Low	Sodium calcium edetate, Tromethamine
Iohexol	Omnipaque 350	Non-ionic	Monomer	350	884	Low	Sodium calcium edetate, Tromethamine
Ioversol	Optiray 300	Non-ionic	Monomer	300	651	Low	Sodium calcium edetate, Tromethamine
Iopamidol	Isovue-370	Non-ionic	Monomer	370	796	Low	Sodium calcium edetate, Tromethamine
Iobitridol	Xenetix 350	Non-ionic	Monomer	350	915	Low	Sodium calcium edetate, Tromethamine
Ioxilan	Oxilan 350	Non-ionic	Monomer	350	695	Low	Sodium calcium edetate, Tromethamine
Iomeprol	Iomeron 350	Non-ionic	Monomer	350	618	Low	Tromethamine
Iopentol	Imagopaque 300	Non-ionic	Monomer	300	640	Low	Sodium calcium edetate, Tromethamine
Iodixanol	Visipaque 320	Non-ionic	Dimer	320	290	Iso	Sodium calcium edetate, Tromethamine
Iotrolan	Isovist 300	Non-ionic	Dimer	300	291	Iso	Sodium calcium edetate

**Table 2 biomedicines-10-01036-t002:** Categories of acute adverse reactions adapted from the ACR Manual on Contrast Media and ESUR Guideline.

	Allergic-Like/Hypersensitivity	Chemo-Toxic
Mild		
	Limited urticaria/pruritis	Limited nausea/vomiting limited
	Cutaneous edema	Transient flushing/warmth/chills
	Limited “itchy”/”scratchy” throat	Headache/dizziness/anxiety/altered taste
	Nasal congestion	Mild hypertension
	Sneezing/conjunctivitis/rhinorrhea	Vasovagal reaction that resolves spontaneously
Moderate		
	Diffuse urticaria/pruritis	Protracted nausea/vomiting
	Diffuse erythema, stable vital signs	Hypertensive urgency
	Facial edema without dyspnea	Isolated chest pain
	Throat tightness or hoarseness without dyspnea	Vasovagal reaction that requires and is responsive to treatment
	Wheezing/bronchospasm, mild or no hypoxia	
Severe		
	Diffuse edema, or facial edema with dyspnea	Vasovagal reaction resistant to treatment
	Diffuse erythema with hypotension	Arrhythmia
	Laryngeal edema with stridor and/or hypoxia	Convulsions, seizures
	Wheezing/bronchospasm, significant hypoxia	Hypertensive emergency
	Anaphylactic shock (hypotension + tachycardia)	

**Table 3 biomedicines-10-01036-t003:** Positivity rate of various kinds of in vivo and in vitro tests for hypersensitivity of ICM.

Category	Test	Percentage [References]
IHR	Sensitivity of skin test	4.2 to 73% (correlate with the severity of the phenotype) [42,43,44,45,46,47,48,49]
	Specificity of SPT	94.6% [47,48]
	Specificity of IDT	91.4–96.3% [47,48]
	Pooled per-patient positivity rates of skin tests	17% (95% CI, 10–26%) [46]
	Severe IHR-pooled per-patient positivity rates of skin tests	52% (95% CI, 31–72%) [46]
	Negative predictive value of skin test	93% (95% CI, 86–96%) [46]
	Cross-reactivity in skin test	68% (95% CI, 48–83%) [47,48]
	Sensitivity of BAT	46–63% [40,41,42]
	Specificity of BAT	89–100% [40,41,42]
NIHR	Sensitivity of skin test	72% [44,50,51]
	Specificity of skin test	96% [44,50,51]
	Pooled per-patient positivity rates of skin tests	26% (95% CI, 15–41%) [46]
	Pooled per-patient positivity rates of SPT	7% (95% CI, 1–30%) [46]
	Pooled per-patient positivity rates of IDT	22% (95% CT, 13–34%) [46]
	Pooled per-patient positivity rates of patch test	16% (95% CI, 15–41%) [46]
	Cross-reactivity in skin test	39% (95% CI, 29–50%) [47,48]
	Sensitivity of LTT	13 to 75% [61]
Skin test for alternative ICM	Negative predictive value for IHR	94.2% (95% CI, 89.6% to 97.2%) [53]
	Negative predictive value for NIHR	86.1% (95% CI, 72.1–94.7%) [53]

Abbreviations: BAT, basophil activation test; ICM, iodinated contrast media; IDT, intradermal test; IHR, immediate hypersensitivity reaction; LTT, lymphocyte transformation test; NIHR, non-immediate hypersensitivity reaction; SPT, skin prick test.

## Data Availability

Not applicable.

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
