# Peer review of "Hypersensitivity Reactions to Iodinated Contrast Media"

_biomedicines, 2022, doi:10.3390/biomedicines10051036_

Round 1

Reviewer 1 Report

Very well developed review. The tables are excellent, and the bibliography very detailed. I only find a deficit on future (already present) therapeutic approaches such as talking about desensitization (if this is feasible) as well as the new pretreatment guidelines used in chemotherapy (see various works by DR MC Castells, from Harvard University )

Author Response

Very well developed review. The tables are excellent, and the bibliography very detailed. I only find a deficit on future (already present) therapeutic approaches such as talking about desensitization (if this is feasible) as well as the new pretreatment guidelines used in chemotherapy (see various works by DR MC Castells, from Harvard University )

Reply:

Thanks for your precious suggestion and we have added more details about therapeutic approaches on page 9, paragraph 8 as “Prescreening, prevention and rapid drug desensitization” “In patients with a hypersensitivity history to their first-choice treatment, rapid drug desensitization is a therapeutic approach by temporarily building tolerance to the drug in a short period of time. The principle of desensitization is to gradually eliminate the IgE activation of mast cells through small doses of drugs. This is done by starting with a small dose of the drug and slowly increasing the dose until achieving the therapeutic dose we need or the maximum tolerated dose. Rapid drug desensitization has been practiced for many years in taxanes for cancer patients and biologics such as rituximab, and there is a great clinical need. With rapid drug desensitization, the chance of breakthrough reaction is usually less than 10% and is usually a mild reaction. Premedication is also included in some desensitization protocols. Despite the long-existing therapeutic approach, there is no evidence to support such drug desensitization in radiocontrast agent hypersensitivity.

Reviewer 2 Report

The paper is a review on the management of hypersensitivity reactions to iodinated contrast media.

The paper is well written and provides practical information to clinicians.

In my opinion, more information should be added:

  • page 6, paragraph 6: I think skin test should be performed in all patients with a history of hypersensitivity reactions, irrespective of the severity. Skin test may be positive also in case of mild reactions if performed within maximum 6 months from the reaction;
  • page 9, pargraph 7: please add some more details on the treatment of acute life-threatening reactions. Oxygen and corticosteroids might be administered in case of severe immediate reactions;
  • page 10, paragraph 9: information on cross-reactivity among iodinated contrast media should be provided to give the clinician the possibility to choose another safe molecule.

Author Response

page 6, paragraph 6: I think skin test should be performed in all patients with a history of hypersensitivity reactions, irrespective of the severity. Skin test may be positive also in case of mild reactions if performed within maximum 6 months from the reaction;

reply:

Thanks for your precious suggestion and we have modified this paragraph as “.... Although the sensitivity of in vivo and in vitro tests of ICM allergy correlate with the severity of the hypersensitivity [42], skin test may be positive in cases of mild reactions if performed within maximum 6 months from the reaction. Hence, skin test could be performed in all patients with a history of hypersensitivity reactions, if feasible. …”

page 9, pargraph 7: please add some more details on the treatment of acute life-threatening reactions. Oxygen and corticosteroids might be administered in case of severe immediate reactions;

reply: Thanks for your precious suggestion.

“corticosteroids might be administered in case of severe immediate reactions” had already been written in this paragraph as “…for life-threatening anaphylaxis, intramuscular epinephrine is necessary,…”

We have modified this paragraph as “…and adequate oxygen supply is critical.”

page 10, paragraph 9: information on cross-reactivity among iodinated contrast media should be provided to give the clinician the possibility to choose another safe molecule.

reply: Thanks for your precious suggestion.

In “Recommendations” section, second paragraph, alternative ICM had been discussed. To express a clearer meaning, we add “the American College of Radiology recommends changing an alternative ICM in patients with a prior allergic-like reaction” in the second paragraph.

Reviewer 3 Report

Dear Authors,

I read the article with big interest. In particular, I greatly appreciated the attention paid to the role of excipients and the section “Treatment”. Before the publication, I would like to ask you to consider following comments:

Abstract:
Please replace “has low osmolality” with “have low osmolality”, “is safe” with “are safe”.

Text
2. Classification of ICM
Please add appropriate references when classifying means.
Pleace replace "character" with characteristics".

Table 1
Pleace replace "character" with characteristics" in the title.
Please correct "Iompeprol".

3. Classification of ADRs of ICM
ADRs related to ICM are usually divided into two categories: hypersensitivity and chemo-toxic reactions. I suggest to replace “physiological events” with chemo-toxic reactions. This differentiation emphasizes the non-allergic and toxic character of the latter category. Please refer to doi: 10.18176/jiaci.0058.

I suggest not using the term "physiological reaction". Please check the entire manuscript.

4. Epidemiology
“but some are not totally conclusive, including…” please explain the meaning.

Please add these references: doi: 10.23822/EurAnnACI.1764-1489.225, doi: 10.3390/biomedicines10040759.

Figure 1
Please add “with immediate reading” to “skin test” in the upper left of the figure.

Figure 2
Please add the source/references of showed algorithm, or specify if it is Authors’ proposal.
As source, I suggest doi: 10.1186/s12948-020-00128-3, doi: 10.1111/j.1398-9995.2005.00745.x.

6. Diagnosis
“… but the skin test is usually performed long after the hypersensitivity event, which decreases the positivity rate [45, 51, 52]”, please add doi: 10.3390/biomedicines10040759 as a further reference.

Please expand the paragraph on excipients by adding allergy investigations to search for hypersensitivity reactions to these substances.

Kind regards

Author Response

Abstract:
Please replace “has low osmolality” with “have low osmolality”, “is safe” with “are safe”.

Reply:

Thanks for reviewer’s suggestion. The sentences have been modified as “Nowadays, iodinated contrast media (ICM) area mostly non-ionic, have low osmolality, and are safe”. 

Text
2. Classification of ICM
Please add appropriate references when classifying means.
Please replace "character" with characteristics".

Reply:

Thanks for reviewer’s comment. The reference [7] has been added to the end of the first sentence in the “Classification of ICM” section. The “character” has been replaced with “characteristics” and the sentence has been modified as “ Due to its non-ionic characteristics, the osmolality is decreased to 2-3 times that of normal serum, but its radiopacity remains similar”. 

Table 1
Pleace replace "character" with characteristics" in the title.
Please correct "Iompeprol".

Reply:

Thanks for reviewer’s comment. The two words in the Table 1 has been replaced with “ characteristics” and “Iomeprol”. 

3. Classification of ADRs of ICM
ADRs related to ICM are usually divided into two categories: hypersensitivity and chemo-toxic reactions. I suggest to replace “physiological events” with chemo-toxic reactions. This differentiation emphasizes the non-allergic and toxic character of the latter category. Please refer to doi: 10.18176/jiaci.0058.

I suggest not using the term "physiological reaction". Please check the entire manuscript.

Reply:

Thanks for reviewer’s comment. The physiologic events have been replaced with chemo-toxic reactions in the entire manuscript.  

4. Epidemiology
“but some are not totally conclusive, including…” please explain the meaning.

Reply:

Thanks for review’s comment. For clearer meaning, we remove this sentence and modify the sentence as “The potential risks of ADRs are allergy, asthma, cardiac status, anxiety, age, and gender. No external warming of ICM, the injection dose, injection speed, and family history of hypersensitivity to ICM are also risk factors. And the reference doi: 10.23822/EurAnnACI.1764-1489.225 has been added.

Figure 1
Please add “with immediate reading” to “skin test” in the upper left of the figure.

Reply:

Thanks for review’s comment. “with immediate reading” has been added in the Figure 1.

Figure 2
Please add the source/references of showed algorithm, or specify if it is Authors’ proposal.
As source, I suggest doi: 10.1186/s12948-020-00128-3, doi: 10.1111/j.1398-9995.2005.00745.x.
Reply:

Thanks for your precious suggestion. We have made modification as “Figure 2. Algorithm for classification and recommendations for ADRs of ICM. For excipient-induced hypersensitivity, investigations can be performed as in IHR work-up. The algorithm is authors’proposal and modified with references.

  1. Diagnosis
    “… but the skin test is usually performed long after the hypersensitivity event, which decreases the positivity rate [45, 51, 52]”, please add doi: 10.3390/biomedicines10040759 as a further reference.
    Reply:

Thanks for your precious suggestion. We have added the reference “Nucera, E., et al. (2022). "Contrast Medium Hypersensitivity: A Large Italian Study with Long-Term Follow-Up." Biomedicines 10(4).”

Please expand the paragraph on excipients by adding allergy investigations to search for hypersensitivity reactions to these substances.

Reply:

Thanks for your precious suggestion. We have modified the paragraph with “…Hypersensitivity reactions to excipients are immediate, hence, BAT is performed to identify if excipient or ICM per se is the real culprit….”
